# *Artemisia afra* and *Artemisia annua* Extracts Have Bactericidal Activity against *Mycobacterium tuberculosis* in Physiologically Relevant Carbon Sources and Hypoxia

**DOI:** 10.3390/pathogens12020227

**Published:** 2023-02-01

**Authors:** Bushra Hafeez Kiani, Maria Natalia Alonso, Pamela J. Weathers, Scarlet S. Shell

**Affiliations:** Department of Biology and Biotechnology, Worcester Polytechnic Institute, Worcester, MA 01609, USA

**Keywords:** *Artemisia annua*, *Artemisia afra*, artemisinin, *Mycobacterium tuberculosis*, hypoxia, carbon source, cholesterol

## Abstract

*Mycobacterium tuberculosis* (Mtb) is a deadly pathogen and causative agent of human tuberculosis, causing ~1.5 million deaths every year. The increasing drug resistance of this pathogen necessitates novel and improved treatment strategies. A crucial aspect of the host–pathogen interaction is bacterial nutrition. In this study, *Artemisia annua* and *Artemisia afra* dichloromethane extracts were tested for bactericidal activity against Mtb strain mc^2^6230 under hypoxia and various infection-associated carbon sources (glycerol, glucose, and cholesterol). Both extracts showed significant bactericidal activity against Mtb, regardless of carbon source. Based on killing curves, *A. afra* showed the most consistent bactericidal activity against Mtb for all tested carbon sources, whereas *A. annua* showed the highest bactericidal activity in 7H9 minimal media with glycerol. Both extracts retained their bactericidal activity against Mtb under hypoxic conditions. Further investigations are required to determine the mechanism of action of these extracts and identify their active constituent compounds.

## 1. Introduction

Tuberculosis (TB), one of the major fatal diseases of humanity, still poses a major health, social, and economic burden worldwide and mainly occurs in low- and middle-income countries. TB remains a major worldwide health challenge, even 130 years after the discovery of the causative agent *Mycobacterium tuberculosis* (Mtb), causing 10.6 million reported cases and 1.6 million deaths in 2021 [1].

A major challenge of TB is that standard treatment regimens require six months of multi-drug therapy. Drug-resistant TB requires significantly longer treatment regimens, with additionally incapacitating side effects and diminishing results with a limited number of second-line drugs including linezolid, bedaquiline, delamanid, and pretomanid [2,3,4,5]. One contributor to the persistence of TB is the formation of hypoxic granulomas, where Mtb can survive for long periods of time in non-growing states in which it is phenotypically tolerant to most drugs [6,7]. The mechanisms by which Mtb can persist in the host are complex and likely include both low drug penetration and reduced drug efficacy within the granuloma. Immune stressors and the granuloma structure produce populations of Mtb in slow or non-growing states or with altered physiology, causing reduced sensitivity to most antibiotics, termed tolerance. Given this drug tolerance, treatment with the first-line TB antibiotics leads to a significant decrease in viable bacteria, but often fails to sterilize [8,9]. Surviving bacteria likely provide opportunities for the acquisition of genetically encoded drug resistance. The emergence of drug resistance demands the discovery of new drugs and combinations to improve TB therapy at all stages of the disease.

The development of novel therapies is linked with an understanding of different survival strategies by bacterial pathogens to overcome stress. Bacterial nutrition is a key component of host–pathogen interaction in TB [10,11] and, thus, is useful in drug development. The infection pattern of bacterial pathogens and the interaction between drugs and Mtb nutritional requirements may greatly affect disease progression. Lovewell et al. [12] previously reported that Mtb uses lipids of the host stored in functional lipid bodies during infection and intracellular replication, suggesting the importance of lipids during infection. Although lipids are Mtb’s major carbon source in vivo, their low solubility and the presence of multiple carbon sources led to the consideration that Mtb may use additional carbon sources during infection [13,14,15]. Different carbon sources are co-catabolized by Mtb in vitro [15,16,17], demonstrating that Mtb can use multiple carbon sources including glucose, glycerol, and cholesterol. Recent work [18] confirmed that, during infection, mycobacteria use multiple carbon sources. 

Plant extracts have been used for treating various diseases for millennia and about 100,000 plant species have medicinal value [19,20], often without any adverse side effects on human health and the environment [21]. Plant specialty molecules (previously termed plant secondary metabolites; PSM) can affect microbial cells in several ways, including the disruption of membrane function and structure, the interruption of DNA/RNA synthesis and function, interference with intermediary metabolism, the induction of coagulation of cytoplasmic constituents, and the interruption of normal cell communication [22,23,24,25,26]. Antibiotics currently used as therapies to treat bacterial diseases rely on various mechanisms of bacterial growth inhibition and bacterial killing. The development of novel therapies using plants requires a better understanding of the mechanisms of the antimicrobial activity of plant compounds [27].

Many species of the genus *Artemisia* (Figure 1) have different proven medicinal properties and are used for treating diseases such as malaria, cancer, and hepatitis [28,29,30], and have even shown efficacy against COVID-19 [31,32,33,34]. Traditional treatments led to the testing of different *Artemisia* species and their extracts against several pathogens, including mycobacteria in vitro and in vivo in a murine model of tuberculosis [35,36]. *Artemisia annua* and *Artemisia afra* are used globally to treat fever and cough, which are the common symptoms of many diseases including TB [37]. *A. annua* produces the antimalarial drug artemisinin (ART), which also has antitubercular activity [38,39,40]. Both ART and its derivative artesunate were observed to be effective against Mtb in vitro as well as in a rat-infected model [40]. In another study, a mycobactin–artemisinin conjugate was designed which exhibited activity against sensitive as well as drug-resistant strains of mycobacteria at a much lower concentration. The conjugate promoted the initiation of radical reactions, which were predicted to be central to the mechanism of action [41]. ART also targets a key survival pathway for Mtb during non-replicating persistence, by blocking the two-component regulatory system DosRST under hypoxic conditions in vitro [39,42]. Although it produces little to no ART, *A. afra* also had significant bactericidal activity against Mtb [38,43,44].

*A. annua* and *A. afra* dichloromethane (DCM) extracts were previously shown to have strong bactericidal activity against *M. tuberculosis* [38], suggesting that these plants contain bactericidal compounds beyond ART. To our knowledge, however, there are no reports regarding the efficacy of *A. annua* or *A. afra* against Mtb in the presence of different carbon sources and under hypoxia. In the present study, the efficacy of extracts of *A. annua* and *A. afra* against Mtb was measured in different carbon sources and under hypoxic conditions to evaluate the potential of these plant extracts or their constituent compounds for the treatment of TB.

## 2. Methodology

### 2.1. Plant Materials

Dried leaves of *Artemisia annua* L. cv. SAM (MASS 317314) were harvested from plants field-grown in Stow, MA from rooted cuttings as described [45]. Dried leaves of *Artemisia afra* Jacq. ex Willd. cv. MAL (originated from Malawi; FTG181107; batch B#1RbA.10.12.20) were obtained from Atelier Temenos LLC in Homestead, FL. The vouchers for other cultivars of *A. afra* are SEN (LG0019529), LUX (MNHNL17730), and PAR (LG0019528). 

### 2.2. Preparation of Plant Extracts

Dried plant leaves were processed as previously described to produce DCM extracts [38]. DCM extracts were pooled and dried under N_2_ as previously detailed [46], and ART was analyzed using GC-MS as described [47]. Hot water tea infusions were also prepared as detailed in Desrosiers et al. [46].

### 2.3. Bacterial Strains and Culture Conditions

*Mycobacterium tuberculosis* (Mtb) strain mc^2^6230 (Δ*panCD*, Δ*RD1* [48]) was grown at 37 °C and 200 rpm in Middlebrooks 7H9 broth, supplemented with 10% OADC (0.5 g/L oleic acid, 50 g/L bovine serum albumin fraction V, 20 g/L glucose, 8.5 g/L sodium chloride, and 40 mg/L catalase), 0.2% glycerol, 0.05% Tween 80, and 24 μg/mL *pantothenate*. Where specified, Mtb was grown in minimal media (0.5 g/L asparagine, 1 g/L KH_2_PO_4_, 2.5 g/L Na_2_HPO_4_, 50 mg/L ferric ammonium citrate, 0.5 g/L MgSO_4_·7H_2_O, 0.5 mg/L CaCl_2_, and 0.1 mg/L ZnSO_4_) supplemented with 0.1% tyloxapol, 24 μg/mL pantothenate, and 0.1% glycerol or 0.1% glucose or 0.1% cholesterol. Middlebrook 7H10 supplemented with 0.5% glycerol, OADC, and 24 μg/mL pantothenate was used to grow Mtb on solid media. When cultures were plated for CFU, they were first serially diluted by a series of six 10-fold dilutions. Multiple dilutions were plated, and colonies were counted for dilutions that yielded multiple colonies, spaced apart well enough for high-confidence counting.

### 2.4. Determination of the Minimum Inhibitory Concentration (MIC) 

We previously reported MICs of Mtb strain mc^2^6230 for DCM extracts of *A. annua* cv. SAM and *A. afra* cv. SEN using a resazurin microtiter assay (REMA) [38]. Using the same method, we also measured the MIC for *A. afra* cv. MAL and remeasured *A. afra* cv. SEN as well as the water extracts (tea infusions) of *A. annua* cv. SAM. For both MICs and killing assays, Artemisia extract concentrations are expressed as the leaf dry mass from which the extract was obtained. In each case, the DCM extract obtained from 2.94 g of dry leaf mass was resuspended in 1 mL DMSO and the concentration of the resulting solution was expressed as 2.94 g/mL.

### 2.5. A. afra and A. annua Bactericidal Activity against Mtb in Different Carbon Sources

Mtb was grown in 7H9 media with OADC and glycerol as described above and when reaching OD_600_ = 0.8, cells were centrifuged, washed twice with PBS, and resuspended to OD_600_ = 0.8 in 7H9 with OADC and glycerol, or in minimal media containing 0.1% glycerol, 0.1% glucose, or 0.1% cholesterol. After 48 h, each culture was adjusted to a final OD_600_ of 0.1. Mtb suspensions were inoculated in triplicate with a final concentration of extract two times the MIC values, as determined for *A. annua* cv. SAM (MIC = extract from 4.5 mg dried leaves/mL, 2× MIC = extract from 9 mg dried leaves/mL) or *A. afra* cv. MAL (MIC = extract from 2.5 dried leaves/mL, 2× MIC = extract from 5 mg dried leaves/mL). Controls consisting of media with Mtb without added extracts were included, also in triplicate. In all cases, the final volume in each conical tube was 5 mL. Mtb cultures were incubated, shaking at 250 rpm and 37 °C. Mtb samples were taken from each liquid culture at days 0, 2, 4, 6 and 10 of incubation and then serially diluted and plated on 7H10 solid medium in the absence of plant extract. After 15–20 days of growth on 7H10 solid medium in the absence of extract, the resulting Mtb colonies were counted and the CFU/mL was calculated. CFU relative to day 0 were calculated as (CFU from culture on day x)/(CFU from culture on day 0).

### 2.6. A. afra and A. annua Bactericidal Activity against Mtb under Hypoxia

Mtb was grown in 7H9 broth as above, but in this case supplemented with 10% ADC (50 g/L bovine serum albumin fraction V, 20 g/L glucose, 8.5 g/L sodium chloride, and 40 mg/L catalase) rather than OADC, at 200 rpm at 37 °C. Seed cultures were normalized to an OD_600_ = 0.1 in 17 mL using fresh media without antibiotics in a rubber stopper-sealed serum bottle (28 mL total bottle volume). The bottles were sealed with chlorobutyl rubber lids and aluminum caps, and then cultures were grown at 37 °C and 120 rpm to generate hypoxic conditions, as was previously reported [49]. Methylene blue in separate indicator cultures was used as an indicator of hypoxic conditions, and discoloration consistently occurred after eight days. ODs plateaued at approximately 0.35–0.4. Six days after hypoxia was established (14 days after sealing), cultures were treated with two times the MIC values as determined for *A. annua* cv. SAM (9 mg/mL) or *A. afra* cv. MAL (5 mg/mL) extracts. The extracts were injected aseptically using a 0.3 mm syringe to minimize the introduction of oxygen. The same volume of DMSO was also injected under the same conditions to use as a control. The hypoxic Mtb cultures were incubated with shaking at 120 rpm and 37 °C for 2 or 7 days and finally plated on solid media as previously described. Mtb cultures were also plated 14 days after sealing, before adding the respective drugs. All conditions were tested in triplicate, and colonies were counted after 15–20 days.

### 2.7. Statistical Analysis

Statistics were performed in GraphPad Prism 9.2.0. CFUs in kill curve experiments were compared by ANOVA and Dunnett’s multiple comparisons test. 

## 3. Results

### 3.1. Determination of the Minimum Inhibitory Concentration (MIC)

Various plant extracts were tested for their ability to inhibit the growth of Mtb. Both hot water extracts and DCM extracts of both *Artemisia sp*. showed strong growth inhibitory effects against Mtb (Table 1). MICs were expressed as the dry leaf mass from which the extracts were obtained. Hot water extract MICs ranged from 1.3 to 1.7 mg/mL for the *A. afra* cultivars SEN, PAR and LUX; *A. annua* cv. SAM was 1.9 mg/mL (Table 1). A hot water extract of *A. afra* cv. MAL was not measured. MICs for DCM extracts of *A. afra* cv. SEN ranged from 4.8 to 10 mg/mL in replicate experiments, and the MIC for *A. afra* cv. MAL was consistently 2.5 mg/mL (Table 1). ART content of *A. annua* cv. SAM was 17.08 mg/g dry leaf mass, but was undetectable in *A. afra* cv. MAL. Since, in contrast to SEN, MAL was ART-free and readily available as a clonal source in the US, experiments were continued using the MAL cultivar. Considering that MICs of DCM extracts were not substantially different from the hot water extract MICs, yet allowed extracts to be more concentrated, further experiments were conducted using the DCM extracts. 

### 3.2. A. afra and A. annua Bactericidal Activity against Mtb in Different Carbon Sources

DCM extracts of *A. annua* cv SAM and *A. afra* cv MAL were tested for their bactericidal activities against Mtb strain mc^2^6230 (pantothenate auxotroph and RD1 deletion) in minimal media containing each of three different carbon sources individually: glycerol, glucose, and cholesterol. Cultures grown in 7H9 medium containing a mixture of glucose, glycerol, and oleic acid were tested in parallel. Untreated Mtb grew at equivalent rates and to equivalent final yields in all three single carbon sources, with a trend towards more growth in 7H9, as expected due to the latter having been developed as a medium to support maximal Mtb growth (Figure 2). Both extracts were bactericidal in all carbon sources. However, the extent of bactericidal activity differed according to carbon source in some cases. *A. afra* extracts showed similar bactericidal activity in all of the tested carbon sources. There were only small differences that were inconsistent across time-points in its bactericidal activity when Mtb was grown in glycerol, glucose, or cholesterol, or in 7H9 with both glucose and glycerol (Figure 2). However, the bactericidal activity of the *A. annua* extract was lower when Mtb was grown in cholesterol or glucose in comparison to growth in glycerol or 7H9 with both glucose and glycerol, suggesting different effects of carbon metabolism on the bactericidal activities of the two extracts (Figure 2). 

### 3.3. A. afra and A. annua Bactericidal Activity against Mtb under Hypoxia 

Considering that Mtb survives under hypoxic conditions within lung granulomas during natural infection, *A. annua* cv. SAM and *A. afra* cv. MAL extracts were tested for their bactericidal activity against Mtb strain mc^2^6230 grown under hypoxic conditions. Cultures were grown in 7H9 medium containing a mixture of glucose and glycerol and sealed in bottles with a headspace:culture volume ratio of 0.6 at an OD of 0.1. Extracts and DMSO were added 14 days after sealing the bottles, by injection through the rubber lid to minimize the introduction of oxygen. Over seven days after injection, the viable population of Mtb remained relatively constant in cultures treated with only DMSO. We observed a small increase in the number of viable cells on Day 2 after DMSO treatment, which may be due to minimal oxygen introduction during drug injection. However, the viable Mtb cell population declined by one and three orders of magnitude at 2 and 7 d, respectively, after the injection of *A. afra* or *A. annua* DCM extracts into the hypoxic cultures (Figure 3). It was therefore concluded that *A. afra* and *A. annua* have similar bactericidal effects under hypoxic conditions.

## 4. Discussion

To our knowledge, this is the first study showing that *Artemisia* extracts can kill Mtb regardless of the carbon source used for its growth, and when Mtb is in a hypoxia-induced state of growth arrest. This is important because Mtb metabolizes a variety of different sources of carbon that it likely encounters during infection [15,16,17,50,51,52]. The results in this study showed that growth rates and yields were similar in glucose, glycerol, and cholesterol, and the *A. afra* cv. MAL extract was equally efficacious in all three conditions. However, the *A. annua* cv. SAM extract efficacy differed among carbon sources, suggesting that there may be two different mechanisms of killing action in different carbon sources in response to each *Artemisia sp*.

Plants have a long history of providing bioactive compounds for vital and novel therapeutics [53,54]. Many previous studies reported the effect of *A. annua* and *A. afra* extracts against the growth of Mtb [35,36,38,43,44]. Most prior studies were in vitro; however, Ntutela et al. [44] also included rodent testing. Although those in vitro tests showed a DCM extract MIC of 290 µg/mL against Mtb H37Rv, and a subfraction activity (fraction C8) that at 2 µg/mL was nearly 100 times more potent than the original extract, there was no efficacy of either the DCM or hot water extract in vivo in Mtb-infected mice (Table 1). 

The extraction methods used in Ntutela et al. [44]’s studies may be a reason for the loss of activity in vivo. For example, 200 g of dried leaves were boiled in 4 L water for 30 min, filtered and then freeze dried to yield a hot water infusion. The traditional hot water extraction was boiled for 5–10 min at 5 g/L or steeped for 5–10 min in the boiled water. In our unpublished studies, increasing the dry mass of the *Artemisia*:water ratio beyond 10 g/L results in >50% loss of extractables. Furthermore, increasing the leaf g/L beyond ~10 g/L causes a significant decline in extractables [55]; e.g., ART recovery declined from 62 to 29% when leaf:hot water ratios increased from 20 to 50 g/L. The Ntutela et al. study used a water extract of 50 g/L with 30 min boiling and lyophilized to dryness. In our experience with *A. annua,* ART was not fully recovered when our lyophilized tea infusion was reconstituted in water, resulting in about a 50% loss, so for experiments we never use reconstituted lyophilized tea (Appendix A). When compared to our tea MIC of ~1.5 mg/mL, the Ntutela et al. [44] water extract had neither in vitro nor in vivo activity (Table 1), results consistent with a possible loss of activity of the water extract in vivo. A similar argument can be made for the DCM extract of Ntutela et al. [44], because they extracted at 2 g/10 mL, then twice more but each at half the original solvent volume (Table 1). In contrast, we extracted 1 g/20 mL DCM for 30 min in a sonicating water bath at room temperature and extracted twice again using the same volumes [38]. One might question whether the extracts even reached the lungs of the treated mice; however, an ADME study of rats treated per os with *A. annua* showed that a considerable amount of the marker drug ART reached the lungs [46]. Finally, Ntutela et al. [44] did not gavage their animals with the *Artemisia* extracts, but instead mixed them into the feed for ad libidum consumption, which compromised dosing.

Our results revealed major survival differences of Mtb in different carbon sources when exposed to *A. annua* and *A. afra* extracts. The efficacy of the *A. afra* extract in cholesterol is particularly important because cholesterol plays a pivotal role in the infectivity and virulence of Mtb [56]. Mtb uses cholesterol as a major carbon source during infection and any other compounds or carbon sources that hinder cholesterol metabolism can inhibit Mtb growth, resulting in carbon starvation and metabolic intoxication and subsequently causing an imbalance in the central metabolism [57]. Chang and Guan confirmed that cholesterol and fatty acid are the main carbon sources that Mtb uses during infection [57]. However, some drugs have variable efficacy depending on the carbon source in use. For example, Kalia and collaborators reported that glycerol supplementation interfered with the potency of drugs targeting cytochrome *bc*_1_*:aa*_3_ in mycobacteria [18]. According to the studies of Bellerose et al., 2019 [58], Mtb mutants that lack glycerol catabolism due to variations in homopolymeric regions in the glycerol kinase gene (*glpK*) are associated with drug resistance in clinical isolates and are less susceptible to treatments. Further, they analyzed the efficacy of glycerol metabolism and other carbon sources (fatty acid and cholesterol) in determining the effect of different drugs (INH, RIF, and moxifloxacin) against Mtb. They found that all drugs were almost 50% less efficient in the absence of glycerol catabolism. That study speculated that glycerol significantly increases the efficacy of different drugs and enhances their efficacy. Highlighting that characteristic, a drug development effort encountered promising lead compounds when screening for activity in glycerol, but found they lacked activity against Mtb grown in other carbon sources [59].

Hypoxia and the gradual depletion of oxygen is a key element in granuloma development in human TB and an important consideration in the design of therapeutics useful for treating tuberculosis [60]. Several in vitro models to obtain non-replicating *M. tuberculosis* have been developed and are based on reducing oxygen availability and nutrient starvation. One of the most used methods is the Wayne model, in which non-replicating Mtb is obtained by gradually adapting stirred aerobic cultures to hypoxia through a self-generated oxygen depletion gradient [61]. Our method was an adaptation of the Wayne model.

Several lines of evidence link the inhibition of Mtb growth/metabolism with hypoxic conditions within the host. Mtb can survive long periods of hypoxia but is an obligate aerobe for growth. Tuberculosis infections are preferentially associated with the most oxygen-rich sites in the body [62], suggesting that reduced levels of O_2_ may limit Mtb growth in vivo. This may explain why recrudescent tuberculosis occurs most often in the upper lobes of the lung, the single most-oxygenated region of the body. According to Lim et al. [63], phosphoenolpyruvate (PEP) is almost completely depleted in Mtb under hypoxic conditions. A loss of PEP reduces PEP–carbon flux toward multiple pathways essential for the replication of Mtb. Metabolomic profiles of Mtb collected under hypoxia showed an accumulation of intermediates in glycolysis and the reductive branch of the TCA cycle, with a reciprocal depletion of PEP and oxidative branch intermediates of the TCA cycle, such as α-ketoglutarate. Under hypoxia, PEP depletion may affect multiple cellular metabolic processes that are involved in Mtb metabolic remodeling [51]. 

The impact of hypoxia during TB treatment is likely tremendous, because non-growing Mtb is less sensitive to most drugs. For example, isoniazid (INH) is one among the four first-line drugs used *in* the treatment of *tuberculosis, and* in vitro *kills* actively growing bacilli, but it possesses little or no activity against Mtb under conditions of nutrient starvation or progressive oxygen depletion [64,65]. In contrast, we found that *A. annua* and *A. afra* extracts were bactericidal to Mtb under hypoxia. Thus, these *Artemisia sp*. may maintain efficacy against Mtb in vivo where it is exposed to hypoxic conditions. 

## 5. Conclusion

*Mycobacterium tuberculosis* is a deadly pathogen, causing disease in more than 10 million people annually. In this study, Mtb was killed by *A. annua* and *A. afra* extracts when grown in glycerol, glucose, or cholesterol as sole carbon sources and under hypoxic conditions. Hence, it is important to understand the metabolism of *Mtb* under alternative carbon sources and under hypoxia in the presence of extracts/drugs, given its consequences on infection, antibiotic efficacy, and potential in novel therapeutic development. It is important to unlock the underlying mechanism of how *Artemisia* extracts affect the growth of Mtb in different carbon sources and in hypoxia. This study can serve as a basis for further developing that understanding for future potential use against this multi drug resistant pathogen.

## Figures and Tables

**Figure 1 pathogens-12-00227-f001:**
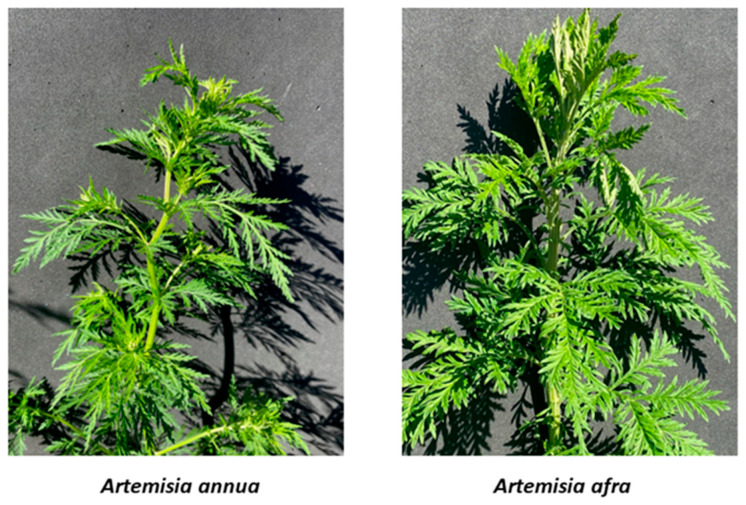
*Artemisia annua* and *Artemisia afra*. Photos courtesy of James Kishlar of Atelier Temenos.

**Figure 2 pathogens-12-00227-f002:**
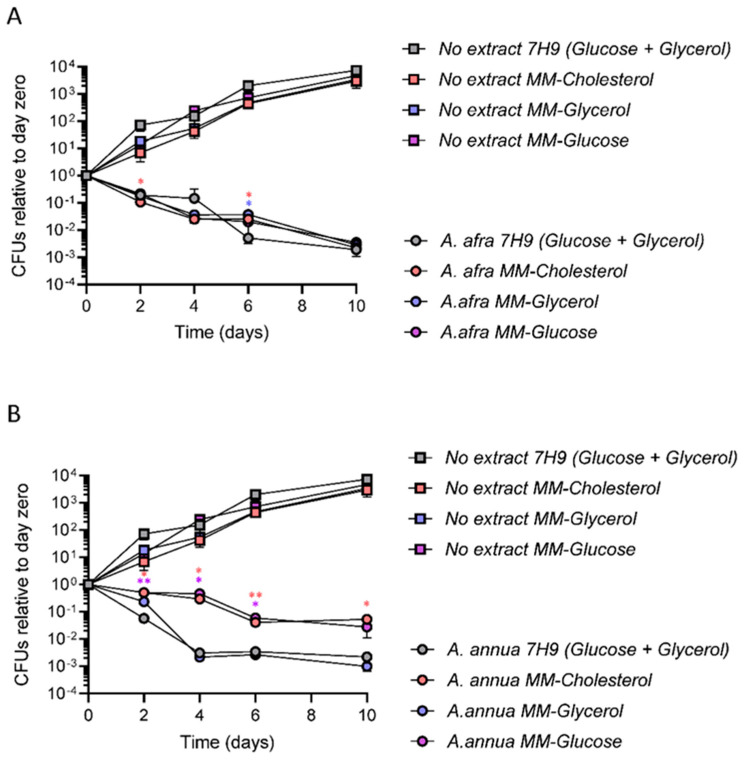
Comparative analysis of Mtb growth and plant extract bactericidal activity in different carbon sources. At day 0, log-phase cultures grown in the indicated carbon sources were back-diluted to OD_600_ 0.1 in the same media and treated with either the indicated extract or with the extract vehicle DMSO. At the time-points indicated on the x axis, aliquots from each culture were removed, diluted, and plated on extract-free solid 7H10 media to determine the number of viable CFU. CFU counts for extract-treated cultures were compared to Day 0 and to each other by 2-way ANOVA. All samples from Day 2 onward were significantly different than Day 0 (adjusted *p* < 0.05). Samples grown in minimal media with individual carbon sources that were significantly different from corresponding samples grown in 7H9 are indicated with stars. * = adjusted *p* < 0.05; ** = adjusted *p* < 0.01. (**A**). Comparative growth/death of Mtb in the presence of 5 mg/mL *A. afra* extracts in all carbon sources. (**B**). Comparative growth/death of Mtb in the presence 9 mg/mL *A. annua* extracts in all carbon sources. The following are the sole carbon sources used in this experiment: 7H9 + glucose and glycerol, minimal medium (MM) + glycerol, MM + glucose, and MM + cholesterol. Data are the average from three independent experiments. Plots showing the growth of Mtb in each individual carbon source in the presence of each plant extract (n = 3) ±SD are provided in the supplementary material (Appendix A).

**Figure 3 pathogens-12-00227-f003:**
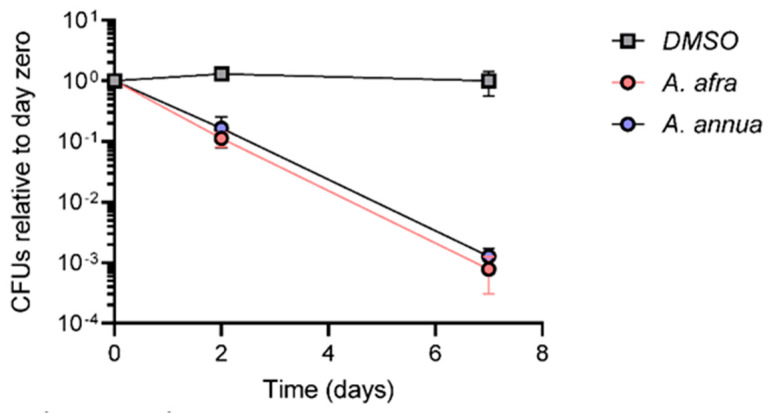
Comparative analysis of *A. afra* and *A. annua* extract bactericidal activities against Mtb under hypoxic conditions. Quantities of 5 mg/mL *A. afra* extract, 9 mg/mL *A. annua* extract, or vehicle control were added by syringe to Mtb cultures that had previously been slowly depleted of oxygen over 14 days. At the time-points indicated on the x axis after adding extracts or vehicle, cultures were opened, diluted, and plated on extract-free 7H10 plates to determine viable CFU. N = 3 cultures per time-point; bars = ±SD. Extract-treated samples at Days 2 and 7 all had significantly lower CFU than samples at time 0 (2-way ANOVA with Dunnett’s multiple comparisons test, *p* < 0.0001). Samples treated with *A. afra* did not differ significantly from samples treated with *A. annua* at any time-point (*p* > 0.05).

**Table 1 pathogens-12-00227-t001:** *A. afra* and *A. annua* comparison of extraction and inhibition data.

Cultivar	Part Tested ^a^	Solvent ^b^	Mtb Inhibition ^c^	Extraction Method	Test System ^d^	Ref. ^e^
** *Artemisia annua* **						
SAM	L	D	4.5 mg/mL	1 g/20 mL DCM, 30 min in sonicating water bath, rt, repeat twice, pooled, dry under N2 5 g/L water, boiled 10 min, cooled, filtered, −20 °C storage	I	This study
		W	1.9 mg/mL		I	
NS	L + St	M	5 mg/mL	1 kg/2.5 L, 72 h rt, repeat twice, pool, rot. evap. 100 g/500 mL, 72 h rt, then vacuum dry	I	[36]
		W	5 mg/mL		I	
NS	L + St	D	77% inhib. by 100 µg/mL extract	500 g/UNK vol solvent, sit at rt 12–48 h	I	[35]
** *Artemisia afra* **						
SEN	L	D	4.8 mg/mL	1 g/20 mL DCM, 30 min in sonicating water bath, rt, repeat twice, pooled, dry under N_2_	I	[38]
SEN	L	D	5–10 mg/mL	1 g/20 mL DCM, 30 min in sonicating water bath, rt, repeat twice, pooled, dry under N_2_ 5 g/L water, boiled 10 min, cooled, filtered, −20 °C storage	I	This study
MAL		D	2.5 mg/mL			
SEN		W	1.3 mg/mL			
PAR		W	1.7 mg/mL			
LUX		W	1.5 mg/mL			
NS	L	D	290 µg/mL	1 kg/10 L, rt 3 h stir, sit overnight, filter, rot. evap., repeat twice w 5 L DCM, pool evap. extracts 200 g/4 L, 30 min boil, filter, freeze dry.	I	[44]
			NA		M	
		W	NA		M	
NS	L	E	NA	50 g/500 mL, 24 h rt, repeat twice, pool, rot. evap.	I	[43]

^a^ L, leaves; St, leaves + stems, in all cases dried and then ground prior to extraction. ^b^ Solvents: D, dichloromethane, E, ethanol; M, methanol; W, water. ^c^ All reported as MICs, if known, otherwise % inhibition. Mass refers to the amount of dry leaf mass used to produce extract, unless otherwise indicated. ^d^ M, in vivo mouse; I, in vitro. ^e^ Results obtained in this study are indicated as such. Results with numbered references were reported in those studies and are reproduced in this table to facilitate comparison of results from different studies. NS, not specified; NM, not measured; NA, no activity; rt, room temperature.

## Data Availability

All data have been provided in the manuscript and its supplemental material.

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
