# Peer review of "Artemisia afra and Artemisia annua Extracts Have Bactericidal Activity against Mycobacterium tuberculosis in Physiologically Relevant Carbon Sources and Hypoxia"

_pathogens, 2023, doi:10.3390/pathogens12020227_

Round 1

Reviewer 1 Report

In this manuscript Bushra Hafeez Kiani et al. provide the important findings of A. annua and A. afra plant extracts strong antimicrobial properties against M.tb pathogen under different carbon sources and under hypoxia. This approach would help to target mycobacterial pathogens and would provide a better therapeutic approach and possible hope in control of emergence of drug resistant pathogens.

The manuscript has a potential finding, the experiments were well designed, and the conclusions are appropriate. I believe that the findings of the manuscript are of sufficient novelty and breadth to merit publication in Pathogens Journal. 

Below are suggestions for minor revision of the manuscript:

1.     Authors need to rewrite the introduction part for example this paragraph below.

“Plant specialty molecules (previously termed 61 plant secondary metabolites; PSM) can affect the microbial cell in several ways including 62 disruption of membrane function and structure, interruption of DNA/RNA synthesis and 63 function, interference with intermediary metabolism, induction of coagulation of cyto-64 plasmic constituents, and interruption of normal cell communication [23-27]. Antibiotics 65 currently used as therapies to treat pathogenic diseases rely on the mechanism of bacterial 66 growth inhibition. Development of novel therapies using plants to requires better under-67 standing of mechanisms of antimicrobial activity of plant compounds.”

2.     Line 147 temperature degree symbol position need to be corrected.

3.     Though authors have reported the antimicrobial potential of A. annua and A. afra plant extracts, it would be nice to see an additional experiment on providing the antimicrobial potential in human macrophage model systems. The additional experiment is not needed if it has already been done by other groups.

However, I have no major concerns with this paper to publish this manuscript after corrections in pathogens general.

Reviewer 2 Report

After reviewing the manuscript entitled “Artemisia afra and Artemisia annua extracts have bactericidal 1 activity against Mycobacterium tuberculosis in physiologically 2 relevant carbon sources and hypoxia”, some modifications must be considered.

The topic of the manuscript is well chosen and argued, but some errors must be resolved before the manuscript is considered suitable for publication.

I would recommend writing the article in third person. Therefore, make changes to eliminate “our” or others … review the entire manuscript and keep consistency.

The images in figure 1 are of low quality.

Please specify the scientific name of the species using the International Code of Nomenclature, adding the authority after the binomial name and the family. It should be named the first time mentioned.

Line 76: There is a link in the word “artesunate”. I don't know if it was intentional

I would number all sections and subsections as they are in the journal template. It is much easier to follow the order of the article.

I would perform statistical analysis on the curves, comparing trials with and without Artemisia species.

In general terms, the manuscript, although a bit simple, presents quality and a good coherence of the arguments. Therefore, in the opinion of this reviewer, the recommendation is to accept after making these suggested minor changes.

Reviewer 3 Report

1- in line 76,, Why did you underline this word  and why did you use a different line size?

2- You list the following in the methodology section:

plant materials

plant extraction

bacterial strain and culture condition

primary screening for antibacterial activity of plant extracts

Determination of the Minimum Inhibitory Concentration (MIC)

A. afra and A. annua bactericidal activity against Mtb under hypoxia

statistical analysis (this is missed )

3- For the first time in the text, any name is written in full. No problem if you want to add it at the end of the manuscript, but you must first write it in the text.

4- In lines 104 to 105, how did you first determine the concentration of ART? Second, this content considers a result, not writing it in material and method.

5- Is this one strain or several strains in line 114?

6- Serial dilutions of extract concentrations should be written in details from... to.... in line 120.

7- In line 124, the extraction method should be written separately in details.

8- In your research, you mentioned bactericidal activity, which is only used for compounds that kill bacteria, but you did not confirm this in any of your experiments. After extract treatments, you should reculture in all treatments until you can determine whether the bacteria gown or not; if the bacteria did not grow, you can say that your extracts have bactericidal activity rather than bacteriostatic activity.

9- in line 126,  2.94 g per what (500 g dry leaf mass or ..., ..., ...

10- in line 134, Where did you get these values, why are these values 9 and 5 mg, and why did your mic results not include these values as table 1?

11- In line 137, the results were obtained at various intervals of 0, 2, 4,.... to 10 days or after 15 to 20 days, respectively.

12- In experiments of the antimicrobial activity, you should use positive control (antibiotic).

13- in line 165, rewrite it to ''4.8, 10 and 2.5 mg/mL''

14- in line 199, title of the table is separated from the text

15- Is this inhibition or MIC in table 1 column Mtb inhbition? I believe it is MICs.

16- Table 1 shows the extraction method and references. This section should be written in detail in methodology and removed from the table.

17- in table 1, M and I in vitro mouse, you did not mentioned in methodology remove it

18- in figure 2, rename, from No drug to extract untreated MM elc.

19- You did not specify how you calculated CFUs relative to day zero in the methodology.

20- Give figures letter-labels rather than top or bottom.

21- These values in lines 236 to 238 differ from the values mentioned in the results section.

22- in line 313, killed or inhibited, you did not confirm from this

23- Check that all references are included in the text.

24- Data in tables and figures should be detailed and easy to understand.

25- You did not discuss the role of carbon sources in increasing the antibacterial activity of the extracts in detail.

26-  Change the term "bactericidal activity" to "antibacterial activity."

Round 2

Reviewer 3 Report

that's ok